# Willingness and eligibility to donate blood under 12-month and 3-month deferral policies among gay, bisexual, and other men who have sex with men in Ontario, Canada

**David J. Brennan**[1]\*, **JP Armstrong**[2], **Maya Kesler**[3], **Tsegaye Bekele**[3], **Nathan J. Lachowsky**[4], **Daniel Grace**[5], **Trevor A. Hart**[5,6], **Rusty Souleymanov**[7], **Barry D. Adam**[8]

1 Factor-Inwentash Faculty of Social Work, University of Toronto, Toronto, Canada, 2 Department of Sociology, York University, Toronto, Canada, 3 Ontario HIV Treatment Network, Toronto, Canada, 4 School of Public Health & Social Policy, University of Victoria, Victoria, Canada, 5 Dalla Lana School of Public Health, University of Toronto, Toronto, Canada, 6 Department of Psychology, Toronto Metropolitan University, Toronto, Canada, 7 Faculty of Social Work, University of Manitoba (Fort Garry Campus), Winnipeg, Canada, 8 Department of Sociology, Anthropology, and Criminology, University of Windsor, Windsor, Canada

\* david.brennan@utoronto.ca

**Data Availability Statement:** The datasets generated and/or analysed during the current study

## Abstract

In Canada, gay, bisexual and other men who have sex with men (GBMSM) are a population that are willing to donate blood, if eligible, but have a history of ineligibility and deferrals due to concerns that their blood poses an increased risk of HIV entering the blood supply. Our objective was to examine the proportion of GBMSM who are willing and eligible to donate under the 12-month deferral policy (implemented in 2016) and the 3-month deferral policy (implemented in 2019). Data for this study comes from the #iCruise study, a mixed cohort study designed to examine sexual health outreach experiences through online services and mobile apps among GBMSM in Ontario. A total of 910 participants were recruited between July 2017 and January 2018. Eligibility criteria include identify as male (cisgender or trans-gender); at least 14 years old; having had sex with a man in the previous year **or** identifying as sexually/romantically attracted to other men **or** identifying as gay, bisexual, queer or two-spirit; and living or working in Ontario **or** having visited Ontario four or more times in the past year. Participants completed a baseline and a follow-up questionnaire. A subset of #iCruise participants (n = 447) further completed this questionnaire. Willingness and eligibility to donate blood were assessed under 12-month and 3-month deferral policies. Of the 447 GBMSM surveyed, 309 (69.1%) reported a general interest in donating blood. 109 (24.4%) GBMSM were willing, 75 (16.7%) were eligible, and 24 (5.4%) were both willing and eligible to donate blood under the 12-month deferral policy. Under the 3-month deferral policy, willingness and eligibility to donate blood increased significantly to 42.3% and 29.3%, respectively. The percent of GBMSM who were both willing and eligible to donate blood also increased significantly to 12.3% under the 3-month deferral policy. The increase in willingness to donate blood varied by age, ethnicity, and geographic residence of participants whereas the increase in eligibility to donate blood varied by education level of participants.

are not publicly available due to privacy or ethical restrictions. The Research Ethics Board of the University of Toronto has reviewed our consent form and confirmed that the consent form does not allow us to share the study data publicly due to privacy and ethical concerns. Therefore, the data are not accessible outside the research team. If access to the data is required, please contact the Research Ethics Board at the University of Toronto regarding this manuscript. Email: ethics. review@utoronto.ca.

**Funding:** This research was funded by the MSM Research Grant Program and a Knowledge Translation Grant from Canadian Blood Services (#508572). The #iCruise study, that this study was embedded within, was funded by a Canadian Health Research Institute's Project Grant (#303157) and the Canadian Foundation for AIDS Research (#497877). The funders had no role in study design, data collection and analysis, decision to publish, or preparation of the manuscript. Additionally, DJB and TAH are supported by Ontario HIV Treatment Network Research Chairs in Gay and Bisexual Men's Health, JA is supported by a Joseph-Armand Bombardier Canada Graduate Scholarship from the Government of Canada, NJL is supported by a Scholar Award from the Michael Smith Foundation for Health Research, and DG is supported by a Canada Research Chair in Sexual and Gender Minority Health.

**Competing interests:** The authors have declared that no competing interests exist.

Under the 3-month deferral policy, GBMSM who were 50 years or older, identified as bisexual or other, had a lower education level, and who were not 'out' to others were more likely to be eligible to donate. GBMSM who reported a general interest in donating blood were more likely to be willing to donate blood under both deferral policies. The most common reason for not being interested in donating blood was the MSM deferral policy itself; many participants interpreted the policy as discriminatory for 'singling out' GBMSM or self-assed themselves as ineligible. Among study participants, both willingness and eligibility to donate blood was significantly higher under the 3-month deferral policy. The results suggest that a time-based reduction to a 3-month deferral policy is impactful but limited. Future research should measure GBMSM's willingness and eligibility under the individual risk-based assessment (to be implemented in 2022).

## Introduction

In Canada, the national blood system is dependent on voluntary donations, so procurement of a safe and sufficient blood supply is crucial. Gay, bisexual and other men who have sex with men (GBMSM) are a population that are willing to donate blood, if eligible, but have a history of ineligibility and deferrals–originating from and upheld by concerns that their blood poses an increased risk of HIV entering the blood supply [1–4].

There is a statistically higher prevalence and incidence of HIV among GBMSM [5–7] and, in Canada, GBMSM's relative risk of contracting HIV is 131 times higher than other men [8], which is also true in its most populous province, Ontario [9]. However, according to the Public Health Agency of Canada (PHAC) an estimated 13% of Canadians living with HIV were unaware of their status [10]. Further, as HIV testing has become more accurate, the "window period" between HIV exposure and HIV detection via blood testing has shortened to an approximately 9-day period via nucleic acid testing, which is the standard method of testing blood donated in Canada [4, 11]. All blood products are subject to rigorous post-donation testing, thereby protecting against transfusion of HIV-containing blood [2, 4–7]. Over the last two decades, many countries have implemented changes to their blood donation policies. In 2020, Brazil and Hungary repealed indefinite bans on blood donation for men who have sex with men (MSM), and Spain (2005), Italy (2001) and Argentina (2015) implemented individual risk-based assessments for blood donation [4, 6]. The UK adopted the "For the Assessment of Individualised Risk" (FAIR) approach in early 2021 [12], also shifting from a time-based deferral policy to an individual risk-based assessment. The USA and Canada, among many other countries have updated their policies by reducing time-based deferrals for MSM [3, 7, 13, 14]. Changes to blood donation policies governing MSM, whether a reduction of time-based policies or a shift from time-based to individual risk-based assessment, do not change the frequency of HIV-positive blood donations [5–7].

In 2013, in Canada, the blood donation deferral period for MSM changed from the deferral of men who reported any sexual contact with another man from 1978 onward, to the deferral of men who reported any sexual contact with another man over the previous five years [15]. In 2016, the deferral period was reduced from five years to one year [15]. In 2019, the deferral period was reduced again, from 12-months to 3-months. The donor questionnaire asks potential male donors: "in the last 3 months, have you had sex with a man?" Although it is not specified in the donor questionnaire, sex is defined as any oral sex or anal sex. The long-standing blanket ineligibility of sexually active MSM has been viewed as discriminatory and creating

barriers to participation in blood donation (including being judged, outed, and treated differently), which oppose the social and personal value GBMSM place on blood donation and may negatively affect their willingness to donate [1–3, 14]. Another policy change request, seeking to remove references to sex or gender of partner from the donor questionnaire and adopt an individual risk-based assessment applied to all donors, was put forward by the blood operator Canadian Blood Services (CBS) to the regulator Health Canada and was accepted in April 2022 [15, 16]. The implementation of this policy in September 2022 will end the blanket deferral of sexually active MSM.

Our objective was to identify the proportion of GBMSM who were willing and eligible to donate under the 12-month deferral policy (implemented in 2016) and the 3-month deferral policy (implemented in 2019). Specifically, we examined if demographic variables were associated with shifts in willingness and eligibility when the deferral period is shortened as well as variation in willingness and eligibility under the 3-month deferral policy. We also reported on written responses provided by study participants to offer insight into why GBMSM eligible under the 3-month MSM deferral policy were not willing to donate or, on the other hand, had not tried to donate despite being willing. These donor policies defer based on sexual behaviour (opposed to sexual identity). As such, when referring to the deferral policies we employ the term MSM. But MSM is not a label participants used to describe themselves. Consequently, we use GBMSM to describe participants as a means of respecting their diverse sexual identities. Further, the term GBMSM better captures participants' attitudes about blood donation as the deferral policy is often interpreted as discriminatory because it stigmatizes GBMSM communities and identities [14].

## Materials and methods

### Study design and recruitment

This study was embedded within a mixed methods cohort study called #iCruise. The objective of the #iCruise study was to examine sexual health outreach experiences through online services and mobile apps among gay, bisexual and other men who have sex with men in Ontario, Canada. Baseline recruitment occurred between July 2017 and January 2018 using advertising on websites, mobile-apps, social media and community-based organizations' email listservs. Eligibility to participate in the #iCruise study included participants identifying as male (cisgender or transgender); at least 14 years old; having had 'any' sex (defined as any contact with another person's genital parts) with a man in the previous year or identifying as sexually/romantically attracted to other men or identifying as gay, bisexual, queer or two-spirit; and living or working in Ontario or having visited Ontario four or more times in the past year. Eligible participants completed a baseline questionnaire, followed by 80% randomization to a weekly online diary arm. All participants (irrespective of having been assigned to participate in the diary arm) then completed a follow-up questionnaire 3 months after the baseline. All questionnaires and diaries were completed online. Detailed methods have been previously described [17].

During the #iCruise follow-up questionnaire, eligible participants were invited to participate in this study. Eligibility for this study included: completing the baseline and follow-up #iCruise questionnaires; self-reported HIV-negative or unknown HIV status; providing the forward sorting address (first three letters/numbers of their postal code) or city of residence; and being 17 years or older (eligible to donate blood). A preamble introduced participants to the study and asked participants if they were willing to be contacted within a 3-month period with more details about participation. Participants who were interested agreed to have their #iCruise data linked to this study to allow for longitudinal data analysis. Three months after

participants completed the #iCruise follow-up questionnaire and agreed to be contacted, this study's informed consent form and questionnaire were completed online (between April 2018 to June 2018). All GBMSM completed this study when Canada's blood donation policy excluded men if they reported sex (oral or anal) with another man within the previous 12 months. After data collection for this study was completed, in 2019, the policy excluding men from donating if they reported sex with another man was reduced from 12 to 3 months.

Informed consent was obtained from all participants prior to data collection. Approval for this study was granted by the Research Ethics Boards of the University of Toronto, University of Windsor, University of Victoria, and Ryerson University. Participants who were 17 years old at the time of data collection were deemed to possess the capacity to consent to this research without approval from a parent or guardian by each Research Ethics Board.

## Data collection

The #iCruise study baseline questionnaire asked about sexual behaviour in the previous six and three months. The #iCruise follow-up questionnaire was administered three months after baseline and asked about sexual behaviour in the previous three months. This study's questionnaire was administered three months after the #iCruise follow-up questionnaire. In total, 12 months of sexual behaviour recall data were available for analyses. All questionnaires were completed online via an emailed link to Qualtrics (Qualtrics, Provo, UT).

This study's questionnaire covered six domains: 1) demographic information, 2) HIV status, STI status, substance use, 3) sexual behaviour, 4) experience with CBS, 5) knowledge of the deferral policy, and 6) willingness to donate under the current and alternative deferral policies. Participants were compensated $15 CAD for completing this study.

## Measures

**Interest in blood donation and willingness and eligibility to donate blood.** Participants were first asked if they had ever had an interest in donating blood in Canada (yes, no, don't remember, prefer not to answer). Written textual responses describing why participants were not interested in donating blood were requested from participants who indicated a lack of interest. Participants were also asked questions about their willingness to donate blood under a 12-month deferral policy (irrespective of their eligibility) and under a then hypothetical 3-month deferral policy (irrespective of their eligibility).

We then assessed participant's eligibility to donate blood under the two policies described above: the 12-month and the 3-month deferral based on self-reported sexual behaviour across the study questionnaires. Participants answered questions about any oral or anal sex in the previous 6 months during the baseline #iCruise questionnaire, in the previous 3 months in the #iCruise follow-up questionnaire (completed 3 months after the #iCruise baseline) and in the previous 3 months in the study questionnaire (completed 3 months after the #iCruise follow-up questionnaire). This allowed for an assessment of sexual behaviour (any oral or anal sex) in the previous 12 months and previous 3 months. If participants reported anal sex (receptive or insertive), they were asked about the HIV status of those partners and condom use frequency if condom use during anal sex was reported.

We were then able to assess participants' eligibility and willingness to donate blood separately and together specifically for the 12-month and 3-month deferral policies. Participants were categorized as willing to donate and then separately as eligible to donate, under the 12-month deferral policy and then under the 3-month deferral policy.

**Socio demographic and other characteristics.** Age at baseline was categorized as 17–29, 30–49 and 50 and older. Sexual orientation was categorized as gay vs. bisexual/other ('other'

included two-spirit, mostly straight, queer, asexual, pan-sexual, questioning, and unsure). Ethnicity was categorized as White vs non-White in some tabulations due to small sample sizes of non-White identified participants. For descriptive purposes, where possible, race/ethnicity was expanded and reported as White, African/Caribbean/Black, East/South East Asian, South Asian, Indigenous, Latino/Brazilian/South American and Other. Marital status was categorized as married/common-law partner vs. other and employment was categorized as 'yes' working full/part time vs. 'no'. Personal annual income was reported in $10,000 CAD increments and grouped into four categories (less than $20,000, $20,000 to $39,999, $40,000 to $59,999 and $60,000 or greater). Education level was reported as: finished high school or less, some post-secondary education, and university degree or higher. The region of Ontario was categorized as: Eastern Ontario, Greater Toronto Area, Northern Ontario, and Southwestern Ontario. Rurality was self-reported as living in a rural/remote area. Participants reported if anyone was aware of their sexual orientation (yes, no/prefer not to answer).

## Statistical analyses

Sociodemographic characteristics of study participants were summarized using descriptive statistics (i.e., frequencies and percentages). We compared the number of participants willing and eligible to donate blood under the 12-month deferral policy with the number of participants willing and eligible to donate blood under the 3-month deferral policy using McNemar Chi-square tests. We also compared these numbers stratified by general interest in donation (yes vs. no/don't remember/prefer not to answer). We then performed a series of cross tabulations to investigate the association between sociodemographic characteristics and the overall increase in the number of participants willing and eligible to donate, from the 12-month deferral policy to the 3-month deferral policy, as well as associations between sociodemographic characteristics and the proportion of participants eligible and willing to donate blood under the 3-month deferral policy. Pearson chi-square or Fisher's exact tests were used to compare willingness and eligibility to donate blood by sociodemographic characteristics of participants. Statistical significance was determined at the $p < 0.05$ level (two-sided test). Statistical analyses were conducted using SAS software version 9.4.

## Textual analysis

An initial analysis of textual data identified a broad range of general and specific reasons of disinterest in blood donation. Similar reasons were classified together to create the codes which were applied to the textual data during a secondary analysis. Written answers with multiple reasons for disinterest were coded separately for each applicable conceptual topic. Coding and creating broad conceptual topics involved collaboration between two members of the research team to ensure intercoder reliability.

## Results

Four hundred and forty-seven eligible GBMSM completed this study. Nearly half of participants were 17 to 29 years old (45.6%), the majority identified as gay (80.5%) and White (61.7%), were not married or common-law (83.0%), were employed full/part time (77.9%), had a personal annual income of $20,000 CAD or more (69.5%), were 'out' about their sexual orientation to at least one person (95.7%), and lived in the greater Toronto area (GTA; 62.4%). Only 13.6% of participants lived in a rural area. Almost 70% (n = 309) of participants had ever been interested in donating blood in Canada and this differed significantly (p = 0.038) by age with significantly lower interest among participants 30 to 49 years (61.9%) compared with participants who were 17 to 29 years (72.1%) or 50 years or older (75.9%). Ever being interested to

donate did not significantly differ (p>0.05) by sexual orientation, ethnicity, marital status, education level, working status, personal income, whether participants were 'out,' rurality, or region of Ontario (see Table 1).

Of the nearly 31% (n = 138) of GBMSM who did not indicate interest in donating blood, most (n = 119) provided written insight into their disinterest. The most common reason was the deferral policy concerning MSM (26.9%). Among this subset of 32, some respondents went into greater detail by linking their disinterest to an understanding of the policy as discriminatory (n = 15) while others explained that they assessed themselves as ineligible due to the limitations placed on their sex practices (n = 8). The second and third reasons most cited by participants overall were fear of needles and blood (26.1%) and miscellaneous health reasons (16.8%), such as a propensity to faint when blood is drawn.

Overall, 16.8% of participants (N = 447) were eligible to donate under the 12-month deferral policy. This number increased significantly (p<0.001) to 29.3% under the 3-month deferral policy (Fig 1). The number of participants who were willing to donate blood was significantly higher (p<0.001) for the shorter deferral policy, from 24.4% for the 12-month deferral policy to 42.3% for the 3-month deferral policy. The number of participants who were both willing and eligible also increased significantly (p<0.001), from 5.4% for the 12-month deferral policy to 12.3% for the 3-month deferral policy.

Regardless of interest in blood donation, participants were significantly more likely to indicate willingness to donate for the 3-month deferral policy than for the 12-month deferral policy (Fig 2). Among those interested in blood donation, an additional 20.1% were willing to donate (from 27.5% to 47.6%) under the 3-month deferral policy (p<0.001). For those not interested in blood donation, an additional 13% were willing to donate (from 17.4% to 30.4%) under the 3-month deferral policy (p = 0.002). Under both the 12-month and 3-month deferral policies participants who were interested in blood donation were significantly more likely to indicate willingness to donate than participants who were not interested in blood donation. Under the 12-month deferral policy, participants interested in blood donation were 10.1% more willing than participants not interested in blood donation (27.5% vs. 17.4%, p<0.001) whereas, under the 3-month deferral policy, participants interested in blood donation are 17.2% more willing than participants not interested in blood donation (47.6% vs. 30.4%, p<0.001).

We examined whether the overall increase in the number of participants willing and eligible to donate, from the 12-month deferral policy to the 3-month deferral policy, varied by demographic characteristics (Table 2). Significant variations in the number of participants willing to donate blood were observed by age (p = 0.004), ethnicity (p = 0.025), and geographic residence (p = 0.036). Specifically, a higher increase in the number of participants willing to donate blood occurred among younger participants (17–29 years), those who identified their race/ethnicity as other than White, and those who lived in the GTA or Southwestern Ontario. There was a significant association between variation in the number of participants eligible to donate blood and education level (p = 0.027). Specifically, a higher increase in the number of participants eligible occurred among participants who had a high school or lower level of education.

We also examined whether the number of participants eligible to donate blood under the 3-month deferral policy varied by demographic characteristics (Table 3). Older (50 years+) participants were more likely to be eligible than younger (17–29 years) participants (p = 0.012). Compared with participants who identified as gay, participants who identified as bisexual or other were more likely to be eligible to donate under the 3-month deferral policy (40.2% vs 26.7%, p = 0.013). A significantly higher percentage of participants with lower education were eligible than those with higher levels of education (p = 0.005), even when adjusting for age. On the other hand, participants who reported that other people are aware of their

**Table 1. Demographic characteristics of participants by interest in donating blood.**

| Demographic characteristics | Have you ever been interested in donating blood in Canada? | | | | | | p-value* |
| --- | --- | --- | --- | --- | --- | --- | --- |
| | Yes | | No/don't remember/prefer not to answer | | Total sample | | |
| | [n = 309] | | [n = 138] | | [N = 447] | | |
| **Age** | | | | | | | **0.038** |
| 17–29 | 147 | (47.6%) | 57 | (41.3%) | 204 | (45.6%) | |
| 30–49 | 99 | (32.0%) | 61 | (44.2%) | 160 | (35.8%) | |
| ≥50 | 63 | (20.4%) | 20 | (14.5%) | 83 | (18.6%) | |
| **Sexual orientation** | | | | | | | 0.631 |
| Gay | 247 | (79.9%) | 113 | (81.9%) | 360 | (80.5%) | |
| Bisexual/other | 62 | (20.1%) | 25 | (18.1%) | 87 | (19.5%) | |
| **Ethnicity** | | | | | | | 0.273 |
| White | 196 | 63.4% | 80 | (58.0%) | 276 | (61.7%) | |
| Non-white | 113 | (36.4%) | 58 | (42.0%) | 171 | (38.3%) | |
| **Ethnicity** | | | | | | | *** |
| African, Caribbean, Black | 24 | (7.8%) | 10 | (7.2%) | 34 | (7.6%) | |
| East Asian/South East Asian | 24 | (7.8%) | 26 | (18.8%) | 50 | (11.2%) | |
| South Asian | 16 | (5.2%) | ** | ** | 21 | (4.7%) | |
| Indigenous | 17 | (5.5%) | ** | ** | 22 | (4.9%) | |
| Latino/Brazilian/South American | 20 | (6.5%) | 9 | (6.5%) | 29 | (6.5%) | |
| White | 196 | (63.4%) | 80 | (58.0%) | 276 | (61.7%) | |
| Other | 12 | (3.9%) | ** | ** | 15 | (3.4%) | |
| **Marital status** | | | | | | | 0.690 |
| Married/common-law partner | 54 | (17.5%) | 22 | (15.9%) | 76 | (17.0%) | |
| Other | 255 | (82.5%) | 116 | (84.1%) | 371 | (83.0%) | |
| **Education level** | | | | | | | 0.746 |
| ≤High school | 30 | (9.7%) | 13 | (9.4%) | 43 | (9.6%) | |
| Some post-secondary education | 132 | (42.7%) | 54 | (39.1%) | 186 | (41.6%) | |
| University degree or higher | 147 | (47.6%) | 71 | (51.4%) | 218 | (48.8%) | |
| **Working FT/PT** | | | | | | | 0.274 |
| Yes | 245 | (79.3%) | 103 | (74.6%) | 348 | (77.9%) | |
| No | 64 | (20.7%) | 35 | (25.4%) | 99 | (22.1%) | |
| **Personal income** | | | | | | | 0.780 |
| <$20,000 | 76 | (24.6%) | 39 | (28.3%) | 115 | (25.7%) | |
| $20,000 - $39,999 | 78 | (25.2%) | 33 | (23.9%) | 111 | (24.8%) | |
| $40,000 - $59,999 | 68 | (22.0%) | 26 | (18.8%) | 94 | (21.0%) | |
| ≥60,000 | 72 | (23.3%) | 34 | (24.6%) | 106 | (23.7%) | |
| Not reported | 15 | (4.9%) | 6 | (4.3%) | 21 | (4.7%) | |
| **Is anybody aware of your sexual orientation?** | | | | | | | 0.112 |
| Yes | 299 | (96.8%) | 129 | (93.5%) | 428 | (95.7%) | |
| No/prefer not to answer | 10 | (3.2%) | 9 | (6.5%) | 19 | (4.3%) | |
| **Live in rural/remote area** | | | | | | | 0.804 |
| Yes | 43 | (13.9%) | 18 | (13.0%) | 61 | (13.6%) | |
| No | 266 | (86.1%) | 120 | (87.0%) | 386 | (86.4%) | |
| **Region of Ontario** | | | | | | | 0.057 |
| Eastern Ontario | 47 | (15.2%) | 11 | (8.0%) | 58 | (13.0%) | |
| Greater Toronto Area (GTA) | 181 | (58.6%) | 98 | (71.0%) | 279 | (62.4%) | |
| Northern Ontario | 19 | (6.1%) | 8 | (5.8%) | 27 | (5.6%) | |

(*Continued*)

**Table 1.** (Continued)

| Demographic characteristics | Have you ever been interested in donating blood in Canada? | | | | | | p-value* |
|---|---|---|---|---|---|---|---|
| | Yes | | No/don't remember/prefer not to answer | | Total sample | | |
| | [n = 309] | | [n = 138] | | [N = 447] | | |
| Southwestern Ontario | 62 | (20.1%) | 21 | (15.2%) | 83 | (18.6%) | |

* P-values from chi-square tests of independence

** Data suppressed due to small cell size

***Statistical tests were not performed due to small cell sizes

^Chi-square test excludes "Not reported" category

sexual orientation were less likely to be eligible than those who said others are not aware of sexual orientation (28.3% vs 52.6%, p = 0.022). Restricting our sample to eligible participants, we examined whether the number of participants willing to donate blood under the 3-month deferral policy varied by demographic characteristics. Among the subgroup of eligible participants, participants who indicated a general interest in blood donation were significantly more likely to indicate a willingness to donate under the 3-month deferral policy than those who did not report interest (50.6% vs 26.1%, p = 0.007).

When we look specifically at the 21 reasons for disinterest in blood donation, provided by a subset of the 34 GBMSM who are eligible to donate under the 3-month deferral policy, but are not interested in donating blood generally nor willing to donate blood under the 3-month deferral policy, the most common reason provided was the policy regarding MSM (28.6%). Conversely, there were 43 GBMSM who were interested in donating blood generally as well as willing and eligible to donate blood under the 3-month deferral policy. Thirty-one of these respondents had not tried to donate or had never donated blood and listed their reasons for not doing so. The most common response, selected by 45.2% (n = 14) of the respondents, was "I don't know if I'm eligible to donate."

Among respondents (N = 446), 38.3% (n = 171) were unsure or did not know that the donor questionnaire had questions specifically about sex between men. Only 7.9% (n = 34) of

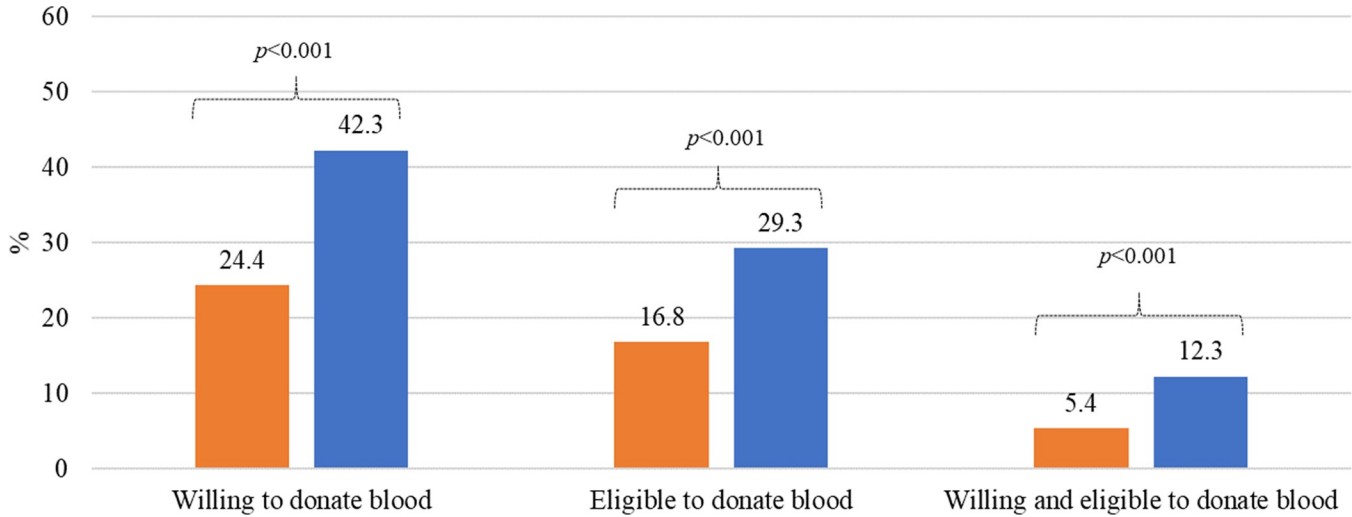

**Fig 1. Willingness and eligibility to donate blood under 12-month and 3-month deferral policies (N = 447).** Orange bar: 12-month deferral policy; Blue bar: 3-month deferral policy. **Note**: Reported p-values are from McNemar chi-square test for paired categorical data.

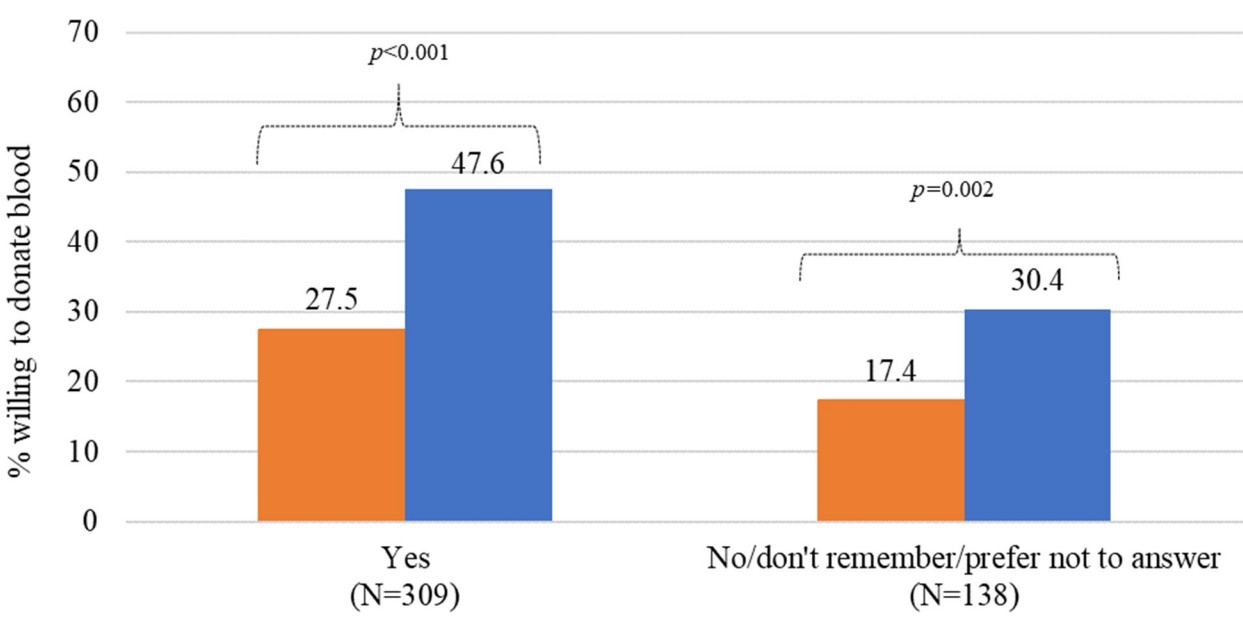

**Fig 2. Willingness to donate blood under 12-month and 3-month deferral policies by general interest in donating blood (N = 447).** Orange bar: 12-month deferral policy; Blue bar: 3-month deferral policy. **Note**: Reported p-values are from McNemar chi-square test for paired categorical data.

participants (N = 432) correctly interpreted the donor questionnaire's operationalization of 'sex' as inclusive of anal sex and oral sex (regardless of ejaculation) but not including rimming, mutual masturbation, or other forms of genital contact.

## Discussion and conclusions

Historically, evidence has suggested that the aging Canadian population, along with the increasing needs for blood, are likely to result in a significant shortage of blood in the near future [18]. Even with approximately 60% of Canadians eligible to donate, less than 4% of eligible Canadians actually donate blood [18]. Nearly 70% of participants in this study sample reported an interest in donating blood at some point, suggesting that GBMSM are a population ready to help combat this potential shortage. However, deferral policies targeting MSM are seen by many as extensive, unscientific, and discriminatory [2–4, 14]. Thus, a general interest in donation may not result in willingness to donate under specific policies if they are interpreted as unjust. Eligibility is also a significant barrier for sexually active GBMSM [7, 14]. We found that, among those in the study sample, willingness and eligibility are significantly higher under the 3-month deferral policy, when compared with the 12-month deferral policy (Fig 1).

There are some important differences by demographic characteristics that should be considered in current and future policy directions related to donor criteria for MSM. GBMSM who reported their race/ethnicity as other than White reported a significantly higher willingness to donate under the 3-month deferral policy. Future research should investigate the reasons behind this substantial increase. However, it is important to note that a binary race/ethnicity measure obfuscates nuance across racialized groups. Indeed, African, Caribbean and Black (ACB) GBMSM report only a nominal increase to willingness (Table 2). This may speak to the other forms of exclusion that ACB GBMSM face as donors such as anti-black racism

**Table 2. Willingness and eligibility to donate blood among study participants under 12-month and 3-month deferral policies.**

| Demographic characteristics | Number of participants | % willing to donate blood | | | % eligible to donate blood | | |
|---|---|---|---|---|---|---|---|
| | | 12-month deferral policy | 3- month deferral policy | p-value | 12-month deferral policy | 3-month deferral policy | p-value* |
| **Age** | | | | **0.004** | | | 0.418 |
| 17–29 | 204 | 21.1% | 46.6% | | 12.3% | 24.5% | |
| 30–49 | 160 | 23.8% | 39.4% | | 17.5% | 28.8% | |
| ≥50 | 83 | 33.7% | 37.4% | | 26.5% | 42.2% | |
| **Sexual orientation** | | | | 0.355 | | | 0.680 |
| Gay | 360 | 22.8% | 41.1% | | 21.8% | 26.7% | |
| Bisexual/other | 87 | 31.0% | 47.1% | | 15.6% | 40.2% | |
| **Ethnicity (binary)** | | | | **0.025** | | | 0.269 |
| White | 276 | 24.6% | 37.0% | | 18.1% | 29.7% | |
| Other | 171 | 24.0% | 50.9% | | 14.6% | 28.7% | |
| **Ethnicity** | | | | *** | | | *** |
| African, Caribbean, Black | 34 | 23.5% | 29.4% | | ** | ** | |
| East Asian/South East Asian | 50 | 26.0% | 56.0% | | 16.0% | 30.0% | |
| South Asian | 21 | 28.6% | 71.4% | | ** | 42.9% | |
| Indigenous | 22 | 27.3% | 45.5% | | ** | 45.5% | |
| Latino/Brazilian/South American | 29 | ** | 55.2% | | ** | ** | |
| White | 276 | 24.6% | 37.0% | | 18.1% | 29.7% | |
| Other | 15 | ** | 53.3% | | ** | 40.0% | |
| **Marital status** | | | | 0.224 | | | 0.842 |
| Married/common-law partner | 76 | 27.6% | 38.2% | | 19.7% | 35.5% | |
| Other | 371 | 23.7% | 43.1% | | 16.2% | 28.0% | |
| **Education level** | | | | 0.662 | | | **0.027** |
| ≤High school | 42 | 31.0% | 45.2% | | 26.2% | 47.6% | |
| Some post-secondary education | 186 | 24.2% | 41.9% | | 22.0% | 31.7% | |
| University degree or higher | 218 | 22.9% | 41.7% | | 10.6% | 23.9% | |
| **Working FT/PT** | | | | 0.989 | | | 0.941 |
| Yes | 348 | 23.9% | 41.4% | | 16.7% | 29.0% | |
| No | 99 | 26.3% | 45.5% | | 17.2% | 30.3% | |
| **Personal income** | | | | 0.641 | | | 0.697 |
| <$20,000 | 115 | 21.7% | 41.7% | | 15.7% | 30.4% | |
| $20,000 - $39,999 | 111 | 26.1% | 39.6% | | 18.9% | 30.6% | |
| $40,000 - $59,999 | 94 | 23.4% | 43.6% | | 18.1% | 35.1% | |
| ≥60,000 | 106 | 28.3% | 45.3% | | 14.2% | 22.6% | |
| **Live in rural/remote area** | | | | 0.423 | | | 0.252 |
| Yes | 61 | 24.6% | 36.1% | | 13.1% | 29.5% | |
| No | 386 | 24.4% | 43.3% | | 17.4% | 29.3% | |
| **Is anybody aware of your sexual orientation?** | | | | 0.364 | | | *** |
| Yes | 428 | 23.8% | 42.1% | | 16.4% | 28.3% | |
| No/prefer not to answer | 19 | 36.8% | 47.4% | | ** | 52.6% | |
| **Have you ever been interested in donating blood in Canada?** | | | | 0.952 | | | 0.902 |
| Yes | 309 | 27.5% | 47.6% | | 15.9% | 27.5% | |
| No/don't remember/prefer not to answer | 138 | 17.4% | 30.4% | | 18.8% | 33.3% | |
| **Region of Ontario** | | | | **0.036** | | | *** |

*(Continued)*

**Table 2.** (Continued)

| Demographic characteristics | Number of participants | % willing to donate blood | | | % eligible to donate blood | | |
|---|---|---|---|---|---|---|---|
| | | 12-month deferral policy | 3- month deferral policy | p-value | 12-month deferral policy | 3-month deferral policy | p-value* |
| Eastern Ontario | 58 | 27.6% | 31.0% | | 22.4% | 27.8% | |
| Greater Toronto Area | 279 | 21.5% | 43.7% | | 15.1% | 26.9% | |
| Northern Ontario | 27 | 36.0% | 37.0% | | ** | 25.9% | |
| Southwestern Ontario | 83 | 28.9% | 47.0% | | 19.3% | 39.8% | |
| Total sample | 447 | 24.4% | 42.3% | <0.001 | 16.8% | 29.3% | <0.001 |

*P-values are from Generalized Estimating Equations (GEE) for categorical variables

**Data suppressed due to small cell size

***Statistical tests were not performed due to small cell sizes

[19]. Previously reported data suggest that, among the general population, older adults are more likely to donate than younger adults, and at a greater frequency [18]. We find that older (50+) GBMSM are significantly more likely than their counterparts to be eligible under the 3-month deferral policy (Table 3). However, despite heightened eligibility, this age group is least likely to indicate a willingness to donate under the 3-month deferral policy (Table 2). Clackett et al. [20]. found similar differences by age among GBMSM in Australia. They suggest that older GBMSM may be less willing to donate after directly experiencing heightened stigma due to the introduction of the MSM deferral policy. Blood operators and regulators should reach out to older GBMSM when considering adjustments to their policies, as a means of better understanding this lack of willingness as well as repairing relationships. GBMSM who identify their sexual orientation as bisexual or other, and GBMSM who are not 'out' (no one else is aware of their sexual orientation), are more likely to be eligible under the 3-month deferral than their respective counterparts (Tables 2 and 3). This finding demonstrates that, compared to other GBMSM, gay men are disproportionately impacted by the MSM deferral policy. Differences in sex behaviour between subgroups within the GBMSM umbrella require further investigation.

This study finds that younger (17–29) and older (50+) GBMSM are similarly interested in blood donation (significantly exceeding the interest levels of GBMSM age 30–49, Table 1). We also find significant differences in willingness between those with and without a general interest in donating blood. Among participants, both those who indicated an interest in blood donation as well as those who did not, willingness to donate blood was significantly higher under the 3-month deferral policy (Fig 2). However, under both the 12-month and 3-month deferral policies, participants who were interested in blood donation were still much more likely to report a willingness to donate (Fig 2). This remains true among those who are eligible to donate under the 3-month variant (Table 3). The results suggest that a time-based reduction to a 3-month deferral policy is impactful for both participant groups, but it is not enough to alleviate the difference in willingness between these two subgroups.

Many of the participants, who did not indicate an interest in donating blood, note the policy as a deterrent–with some specifying that they see the policy as discriminatory. The continued exclusion of MSM from blood donation, even with improved testing and ongoing monitoring and without attention to individual level risk factors, has negatively impacted how GBMSM think and feel about blood donation [1, 4, 21]. The gap in willingness between interested and non-interested participants may persist because a 3-month deferral policy—or any deferral policy singling out MSM—is still be interpreted as unjust and, thus, unacceptable

**Table 3. Characteristics of participants by eligibility and willingness to donate blood under the 3-month deferral policy.**

| Demographic characteristics | Eligible to donate | | Ineligible to donate | | p-value | Among those eligible to donate | | | | p-value |
|---|---|---|---|---|---|---|---|---|---|---|
| | | | | | | Willing to donate | | Not willing to donate | | |
| | [n = 131] | | [n = 316] | | | [n = 55] | | [n = 76] | | |
| **Age** | | | | | 0.012 | | | | | 0.540 |
| 17–29 | 50 | (24.2%) | 154 | (75.5%) | | 24 | (48.0%) | 26 | (52.0%) | |
| 30–49 | 46 | (28.8%) | 114 | (71.3%) | | 18 | (39.1%) | 28 | (60.9%) | |
| ≥50 | 35 | (42.2%) | 48 | (57.8%) | | 13 | (37.1%) | 22 | (62.9%) | |
| **Sexual orientation** | | | | | 0.013 | | | | | 0.356 |
| Gay | 96 | (26.7%) | 264 | (73.3%) | | 38 | (39.6%) | 58 | (60.4%) | |
| Bisexual/other | 35 | (40.2%) | 52 | (59.8%) | | 17 | (48.6%) | 18 | (51.4%) | |
| **Ethnicity** | | | | | 0.812 | | | | | 0.375 |
| White | 82 | (29.7%) | 194 | (70.3%) | | 32 | (39.0%) | 50 | (61.0%) | |
| Non-white | 49 | (28.7%) | 122 | (71.3%) | | 23 | (46.9%) | 26 | (53.1%) | |
| **Ethnicity** | | | | | *** | | | | | *** |
| African, Caribbean, Black | ** | ** | 29 | (85.3%) | | ** | ** | ** | ** | |
| East Asian/South East Asian | 15 | (30.0%) | 35 | (70.0%) | | 8 | (53.3%) | 7 | (46.7%) | |
| South Asian | 9 | (42.9%) | 12 | (57.1%) | | 6 | (66.7%) | ** | ** | |
| Indigenous | 10 | (45.5%) | 12 | (54.5%) | | ** | ** | 8 | (80.0%) | |
| Latino/Brazilian/South American | ** | ** | 25 | (86.2%) | | ** | ** | ** | ** | |
| White | 82 | (29.7%) | 194 | (70.3%) | | 32 | (39.0%) | 50 | (61.0%) | |
| Other | 6 | (40.0%) | 9 | (60.0%) | | ** | ** | ** | ** | |
| **Marital status** | | | | | 0.191 | | | | | 0.559 |
| Married/common-law partner | 27 | (35.5%) | 49 | (64.5%) | | 10 | (37.0%) | 17 | (63.0%) | |
| Other | 104 | (28.0%) | 267 | (72.0%) | | 45 | (43.0%) | 59 | (57.1%) | |
| **Education level** | | | | | 0.005 | | | | | 0.819 |
| ≤HS completion | 20 | (47.6%) | 22 | (52.4%) | | 9 | (45.0%) | 11 | (55.0%) | |
| Some post-secondary | 59 | (31.7%) | 127 | (68.3%) | | 23 | (39.0%) | 36 | (61.0%) | |
| University completion or higher | 52 | (23.9%) | 166 | (76.1%) | | 23 | (44.2%) | 29 | (55.8%) | |
| **Working FT/PT** | | | | | 0.805 | | | | | 0.802 |
| Yes | 101 | (29.0%) | 247 | (71.0%) | | 43 | (42.6%) | 58 | (57.4%) | |
| No | 30 | (30.3%) | 69 | (69.7%) | | 12 | (40.0%) | 18 | (60.0%) | |
| **Personal income\*** | | | | | 0.269 | | | | | 0.988 |
| <$20,000 | 35 | (30.4%) | 80 | (69.6%) | | 15 | (42.9%) | 20 | (57.1%) | |
| $20,000-$39,999) | 34 | (30.6%) | 77 | (69.4%) | | 14 | (41.2%) | 20 | (58.8%) | |
| $40,000-$59,999) | 33 | (35.1%) | 61 | (64.9%) | | 14 | (42.1%) | 19 | (57.6%) | |
| ≥$60,000 | 24 | (22.6%) | 82 | (77.4%) | | 11 | (45.8%) | 13 | (54.2%) | |
| **Live in rural/remote area** | | | | | 0.970 | | | | | *** |
| Yes | 18 | (29.5%) | 43 | (70.5%) | | ** | ** | 13 | (72.2%) | |
| No | 113 | (29.3%) | 273 | (70.7%) | | 50 | (44.2%) | 63 | (55.7%) | |
| **Is anybody aware of your sexual orientation?** | | | | | 0.022 | | | | | *** |
| Yes | 121 | (28.3%) | 307 | (71.7%) | | 50 | (41.3%) | 71 | (58.7%) | |
| No/prefer not to answer | 10 | (52.6%) | 9 | (47.4%) | | ** | ** | ** | ** | |
| **Region of Ontario** | | | | | 0.145 | | | | | *** |
| Eastern Ontario | 16 | (27.6%) | 42 | (72.4%) | | ** | ** | 13 | (81.3%) | |
| Greater Toronto Area (GTA) | 75 | (26.9%) | 204 | (73.1%) | | 34 | (45.3%) | 41 | (54.7%) | |
| Northern Ontario | 7 | (25.9%) | 20 | (74.1%) | | ** | ** | ** | ** | |
| Southwestern Ontario | 33 | (39.8%) | 50 | (60.2%) | | 15 | (45.5%) | 18 | (54.5%) | |

*(Continued)*

**Table 3.** (Continued)

| Demographic characteristics | Eligible to donate | | Ineligible to donate | | p-value | Among those eligible to donate | | | | p-value |
|---|---|---|---|---|---|---|---|---|---|---|
| | | | | | | Willing to donate | | Not willing to donate | | |
| | [n = 131] | | [n = 316] | | | [n = 55] | | [n = 76] | | |
| Ever interested in donating blood | | | | | 0.211 | | | | | **0.007** |
| Yes | 85 | (27.5%) | 224 | (72.5%) | | 43 | (50.6%) | 42 | (49.4%) | |
| No/don't remember/prefer not to answer | 46 | (33.3%) | 92 | (66.7%) | | 12 | (26.1%) | 34 | (73.9%) | |

*Income data missing for 21 participants

**Data suppressed due to small cell size

***Statistical tests were not performed due to small cell sizes

[2, 21]. Indeed, Armstrong et al. [21] found that some Canadian GBMSM would only consider donating under an individual risk-based assessment applied to all donors. However, the continued difference in willingness between interested and non-interested GBMSM might also reflect the salience of other barriers noted by our participants, such as the fear of needles and blood. The extant literature similarly details fear, physical reactions, such as fainting, and a lack of convenience as detractors for some potential donors [22, 23] but reasons for donating such as altruism and self-fulfillment can outweigh these common concerns [3, 14, 24].

Although eligibility increased significantly under the 3-month deferral policy, it is still a minority of GBMSM who are eligible to donate (Fig 1). Blanco et al. find that an individual risk-based assessment can increase the total number of donors while maintaining the safety of the blood supply [25]. Thus, Health Canada's acceptance of an individual risk-based assessment represents a step in the right direction [15, 16]. However, future research will be needed to measure the actual impact of this policy on the eligibility (as well as the willingness) of GBMSM in Canada.

Our results also indicate some ambiguity concerning the donor questionnaire's operationalization of 'sex'. The donor questionnaire asks potential male donors "in the last 3 months, have you had sex with a man?" However, the donor questionnaire does not define what is meant by 'sex' (oral and anal sex, the specific sexual behaviours excluded under the MSM deferral policy). Uncertainty regarding eligibility was a recurrent reason for never having donated, reported by participants who were interested in donating blood as well as willing and eligible to donate under the 3-month deferral policy. Blood operators and regulators should ensure that the future donor deferral criteria is as specific as possible, avoiding blanket terms such as 'sex' which may be interpreted in a wide array of different ways.

## Limitations

The study findings provide insight into changes to willingness and eligibility pertinent to Canada's 2019 time-based reduction to the MSM deferral policy. Because we surveyed participants while the 12-month deferral policy was still in place (and the 3-month deferral was presented to participants as an alternative), we cannot assess the actual impact of this policy change. Our measurements of general interest in blood donation and willingness to donate under the 12-month deferral policy and the (at the time, hypothetical) 3-month deferral policy reflect participants' views while the 12-month deferral policy was still in place.

Eligibility criteria and recruitment for the #iCruise study may have resulted in a more sexually active participant base than the general GBMSM population.

Finally, our use of written responses provides only limited insight into disinterest in blood donation among GBMSM. Only participants who indicated a lack of interest in blood donation were asked to provide reasons why; as such, we cannot speak to the views of participants' who are interested in blood donation but may share their critiques (e.g., seeing the deferral policy for MSM as discriminatory). In-depth qualitative data can better illuminate how to remedy disinterest in blood donation through future policy change (as well as other forms of redress) as a means of remedying this gap in willingness.

## Acknowledgments

We would like to thank our study participants. We would also like to thank Kathryn Wells who contributed to the writing of the manuscript.

## Author Contributions

**Conceptualization:** David J. Brennan, JP Armstrong, Maya Kesler, Nathan J. Lachowsky, Daniel Grace, Trevor A. Hart, Rusty Souleymanov, Barry D. Adam.

**Data curation:** JP Armstrong, Maya Kesler.

**Formal analysis:** David J. Brennan, JP Armstrong, Maya Kesler, Tsegaye Bekele.

**Funding acquisition:** David J. Brennan, JP Armstrong, Nathan J. Lachowsky, Daniel Grace, Rusty Souleymanov, Barry D. Adam.

**Investigation:** David J. Brennan, JP Armstrong, Maya Kesler, Nathan J. Lachowsky, Trevor A. Hart, Barry D. Adam.

**Methodology:** David J. Brennan, JP Armstrong, Maya Kesler, Rusty Souleymanov, Barry D. Adam.

**Project administration:** David J. Brennan, JP Armstrong.

**Resources:** David J. Brennan.

**Software:** JP Armstrong.

**Supervision:** David J. Brennan, JP Armstrong, Maya Kesler, Nathan J. Lachowsky.

**Validation:** Tsegaye Bekele.

**Visualization:** Tsegaye Bekele.

**Writing – original draft:** David J. Brennan, JP Armstrong, Maya Kesler.

**Writing – review & editing:** David J. Brennan, JP Armstrong, Maya Kesler, Tsegaye Bekele, Nathan J. Lachowsky, Daniel Grace, Trevor A. Hart, Rusty Souleymanov, Barry D. Adam.

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
