## [Decision Letter · Decision Letter 0]

9 Jun 2022

PGPH-D-22-00148

Willingness and eligibility to donate blood under 12-month and 3-month deferral policies among gay, bisexual, and other men who have sex with men in Ontario, Canada.

Dear Dr. Brennan,

Thank you for submitting your manuscript to PLOS Global Public Health. After careful consideration, we feel that it has merit but does not fully meet PLOS Global Public Health’s publication criteria as it currently stands. Therefore, we invite you to submit a revised version of the manuscript that addresses the points raised during the review process.

The results of this original research are useful and valuable. As detailed in the reviewers' comments, presentation of results and discussion need to be tighten up. Please submit your revised manuscript by . If you will need more time than this to complete your revisions, please reply to this message or contact the journal office at globalpubhealth@plos.org. Please include the following items when submitting your revised manuscript:

We look forward to receiving your revised manuscript.

Kind regards,

Kevin Escandón, MD, MSc

Academic Editor

Journal Requirements:

1. You indicated that you had ethical approval for your study. In your Methods section, please ensure you have also stated whether you obtained consent from parents or guardians of the minors included in the study or whether the research ethics committee or IRB specifically waived the need for their consent.

- State what role the funders took in the study. If the funders had no role in your study, please state: “The funders had no role in study design, data collection and analysis, decision to publish, or preparation of the manuscript.”

3. Please update the Funding Information in the system to match the details with Financial Disclosure Statement like funders, grant numbers and recipients.

4. Please provide separate figure files in .tif or .eps format.

5. We have amended your Competing Interest statement to comply with journal style. We kindly ask that you double check the statement and let us know if anything is incorrect. 

6. In the online submission form, you indicated that your data will be submitted to a repository upon acceptance.  We strongly recommend all authors deposit their data before acceptance, as the process can be lengthy and hold up publication timelines. Please note that, though access restrictions are acceptable now, your entire data will need to be made freely accessible if your manuscript is accepted for publication. This policy applies to all data except where public deposition would breach compliance with the protocol approved by your research ethics board. If you are unable to adhere to our open data policy, please kindly revise your statement to explain your reasoning and we will seek the editor's input on an exemption. Please be assured that, once you have provided your new statement, the assessment of your exemption will not hold up the peer review process.

Reviewers' comments:

Reviewer's Responses to Questions

**Comments to the Author**

1. Does this manuscript meet PLOS Global Public Health’s publication criteria? Is the manuscript technically sound, and do the data support the conclusions? The manuscript must describe methodologically and ethically rigorous research with conclusions that are appropriately drawn based on the data presented.

Reviewer #1: Yes

Reviewer #2: Yes

2. Has the statistical analysis been performed appropriately and rigorously?

Reviewer #1: Yes

Reviewer #2: Yes

3. Have the authors made all data underlying the findings in their manuscript fully available (please refer to the Data Availability Statement at the start of the manuscript PDF file)?

Reviewer #1: Yes

Reviewer #2: Yes

4. Is the manuscript presented in an intelligible fashion and written in standard English?

Reviewer #1: Yes

Reviewer #2: Yes

5. Review Comments to the Author

Reviewer #1: General Comments:

This study, conducted in Ontario, Canada, assesses the general perception among 447 men who have sex with men of eligibility and willingness to donate blood under 12 and 3 month deferral policies. The authors find the interesting correlations between increased willingness to donate and decreased age, non-White ethnicity, higher education level.

Major Revisions:

- Table 1 shows that the region of Ontario was almost statistically significant. Could the authors briefly note this correlation in the Discussion and speak to which regions are more metropolitan, more diverse, etc.?

Minor Revisions:

- I would strongly advise using MSM instead of GBMSM, as it is not strictly gay and bisexual men only who have sex with other men, as the authors also noted in their Methods section.

- Lines 297-299: “However, regardless of deferral period length, participants interested in blood donation were significantly more likely to indicate willingness to donate than those not interested in blood donation.” – this is a circular argument, I wonder if another conclusion could be made that is more impactful.

- Lines 311-314 are a bit confusing, and unclear what the second part means. Please split into two sentences.

- Lines 327-329: again, this finding seems a bit obvious. Are there any other implications of this increased willingness to donate based on increased interest?

- Lines 377-379 seem a bit too speculative. Do the authors have any evidence, anecdotal or otherwise, to suggest this may be the case?

- What is the significance of having “don’t remember” as one of the qualifiers in Table 1? Was that added because participants stated they did not remember? If not, please omit.

Reviewer #2: This study reports on the willingness and eligibility of a sub-cohort of #iCruise study participants to donate blood under 12- and 3-month deferral policies at Canadian Blood Services. Data were collected from 2018-2019. The larger study focused on sexual health outreach experiences. With 447 responses, participation in this substudy was 49%.

Overall, the study is well-designed and executed and the methodology has been previously published. One could argue that both the larger study participant group and the blood donation questionnaire exhibit self-selection bias. However, blood donors are also a highly self-selected group and this study may inform on those similarities. The statistical analyses are appropriate for the study design.

Outcome responses were not unexpected and the age group differential disinterest may be related to the implementation of the lifetime ban and those maturing under that policy. Would the authors care to speculate on that association?

Also, in light of the new Canadian Blood Service policy removing the eligibility criteria specific to MSM, how do the authors propose to address this?

Specific items:

Page 3, lines 125-128 – the reference to policy implementation will need updating.

6. PLOS authors have the option to publish the peer review history of their article (what does this mean?). If published, this will include your full peer review and any attached files.

**Do you want your identity to be public for this peer review?** For information about this choice, including consent withdrawal, please see our Privacy Policy.

Reviewer #1: No

Reviewer #2: No

---

## [Decision Letter · Decision Letter 1]

18 Nov 2022

Willingness and eligibility to donate blood under 12-month and 3-month deferral policies among gay, bisexual, and other men who have sex with men in Ontario, Canada.

PGPH-D-22-00148R1

Dear Dr. Brennan,

We are pleased to inform you that your manuscript 'Willingness and eligibility to donate blood under 12-month and 3-month deferral policies among gay, bisexual, and other men who have sex with men in Ontario, Canada.' has been provisionally accepted for publication in PLOS Global Public Health.

Best regards,

Julia Robinson

Executive Editor

Reviewer Comments (if any, and for reference):

Reviewer's Responses to Questions

**Comments to the Author**

1. If the authors have adequately addressed your comments raised in a previous round of review and you feel that this manuscript is now acceptable for publication, you may indicate that here to bypass the “Comments to the Author” section, enter your conflict of interest statement in the “Confidential to Editor” section, and submit your "Accept" recommendation.

Reviewer #2: All comments have been addressed

2. Does this manuscript meet PLOS Global Public Health’s publication criteria? Is the manuscript technically sound, and do the data support the conclusions? The manuscript must describe methodologically and ethically rigorous research with conclusions that are appropriately drawn based on the data presented.

Reviewer #2: Yes

3. Has the statistical analysis been performed appropriately and rigorously?

Reviewer #2: Yes

4. Have the authors made all data underlying the findings in their manuscript fully available (please refer to the Data Availability Statement at the start of the manuscript PDF file)?

Reviewer #2: Yes

5. Is the manuscript presented in an intelligible fashion and written in standard English?

Reviewer #2: Yes

6. Review Comments to the Author

Reviewer #2: All comments have been appropriately addressed.

7. PLOS authors have the option to publish the peer review history of their article (what does this mean?). If published, this will include your full peer review and any attached files.

**Do you want your identity to be public for this peer review?** For information about this choice, including consent withdrawal, please see our Privacy Policy.

Reviewer #2: No
